# Influence of frailty in older patients undergoing emergency laparotomy: a UK-based observational study

Kat L Parmar,[1] Lyndsay Pearce,[2] Ian Farrell,[2] Jonathan Hewitt,[3] Susan Moug[4]

## ABSTRACT

**Introduction** The National Emergency Laparotomy Audit (NELA) has reported that older patients (≥65 years) form a large percentage of emergency high-risk cases with increased postoperative morbidity and mortality. With the population continuing to age rapidly, it is clear that a greater understanding of the factors affecting surgical outcomes in older patients is required. Frailty is a relatively new concept taking into account a variety of factors that increase an individual's vulnerability to increased dependency and death. Research has suggested that high frailty scores increase postoperative complications, length of stay and mortality but the majority of these studies have been carried out on elective patients. Knowledge of how frailty affects patients in an emergency setting would aid clinicians' and patients' decision-making process.

**Methods and analysis** This multicentre study will include consecutive adult patients aged 65 years and over undergoing emergency laparotomies over a 3-month period at 52 National Health Service hospitals across the UK. The primary outcome will be 90-day mortality. Secondary outcomes will include length of hospital stay, 30-day complications, change in level of independence and 30-day readmission. This study has been powered to detect a 10% change in mortality associated with frailty (n=500 patients).

**Ethics and dissemination** This study has been approved by the National Health Service Research Ethics Committee. It has been registered centrally with HRA for English sites, NRSPCC for Scottish sites and Health and Care Research Permissions Service for sites in Wales. Dissemination will be via international and national surgical and geriatric conferences. In addition, manuscripts will be prepared following the close of the project.

**Trial registration number** This study is also registered online at www.clinicaltrials.gov (registration number NCT02952430).

[1]Department of General Surgery, Wythenshawe Hospital, Manchester, UK
[2]Department of General Surgery, Manchester Royal Infirmary, Manchester, UK
[3]Division of Population Medicine, Cardiff University, Cardiff, UK
[4]Department of General Surgery, Royal Alexandra Hospital, Paisley, UK

**Correspondence to**
Dr Jonathan Hewitt;
hewittj2@cardiff.ac.uk

## Strengths and limitations of this study

► It is a large-scale multisite study based in the UK.
► Data collated using the established and effective registrar led research networks.
► Frailty collated using the Clinical Frailty Scale, which is quick and simple to use.
► The Clinical Frailty Scale was the only frailty measure collected, a potential limitation.

recommended input for older adults by elderly medicine specialists from findings in their first report, but this was only reported in 10% of all cases.[3] Clinical decision-making in older patients can be difficult as they have the unique challenges of multimorbidity, polypharmacy and cognitive impairment which can occur separately or more commonly in combination. Several risk stratification methods exist to aid the surgical and anaesthetic team, but are limited as they are generally extrapolated from cohorts of much younger patients. A greater understanding of factors involved in surgical outcomes in older patients is therefore required.[4]

Frailty is defined as 'a medical syndrome with multiple causes and contributors that is characterised by diminished strength, endurance and reduced physiological function that increases an individual's vulnerability for developing increased dependency or death'.[5] This definition is now commonplace in Geriatric medicine with frailty routinely assessed in every hospital in the UK with older peoples services.

Assessment of frailty in emergency surgery has been assessed in a limited number of studies. Of those, high frailty scores preoperatively correlate with increased postoperative complications, length of stay, 30-day and 90-day mortality and likelihood of institutionalisation.[6–8] However, there is substantial methodological heterogeneity with few studies focusing solely on older patients, being prospective in design and including

## BACKGROUND

The population is ageing. This has implications for healthcare provision, including surgery.[1 2] The second report of The National Emergency Laparotomy Audit (NELA) in the UK found that over half of patients undergoing major emergency general surgical procedures were older adults (≥65 years) with the highest risk, longest length of stay and highest mortality. NELA had previously

all surgical patients admitted to an acute surgical ward, rather only those undergoing emergency laparotomy. Knowledge of how frailty affects outcomes after emergency laparotomy will aid surgeons in decision-making in this complex group of patients but, most importantly, help to inform the consent process for patients and their families.

### Aims

To assess whether preoperative frailty correlates with outcomes in older surgical patients undergoing emergency laparotomy (Emergency Laparotomy and Frailty (ELF) study).

## METHODS

### Study design

A multicentre observational study.

### Study setting

Hospitals in the UK that provide emergency general surgery have been invited to participate. Fifty-two hospitals have expressed interest in taking part in the audit. Research will be conducted using the established surgical and geriatric registrar-led research networks.[9 10] The methodology for these networks is well described but in brief the networks provide a centrally coordinated research network that promoted and advertised the ELF study. Potential collaborators were invited to take part in data collection, via a standard expression of interest application. The central study team (described below) subsequently provided the ethical approval, protocol, central organisation and long-term delivery of the project. Support was provided by the North West Surgical Trials Centre (www.nwstc.org.uk).

### ELF steering committee

The steering committee comprises surgical trainees and consultant general surgeons, interested in outcomes for older people undergoing surgery. It is formed from two established research groups, the North West Research Collaborative (surgical trainees) and the Older Persons Surgical Outcomes Collaboration (OPSOC; surgeons and geriatricians). The steering committee is responsible for protocol design, data handling, analysis, dissemination of results and the preparation of manuscripts. The ELF steering committee is responsible for the use of data resulting from this project.

### Principal investigators

The principal investigators at each participating site are responsible for organising and leading the local ELF teams. They have submitted relevant documents to local Research and Development departments for approval and ensured that collaborators act in accordance with local clinical governance and guidelines. These local leads act as a link between the local ELF team and the ELF steering committee. They are the first point of contact for local collaborators and are responsible for the dissemination of information to local collaborators from the ELF steering committee.

### Inclusion criteria

► Patients aged ≥65 years.
► Patients who undergo an expedited, urgent or emergency abdominal procedure on the gastrointestinal (GI) tract, including the following:
► Open, laparoscopic or laparoscopic-assisted procedures.
► Procedures involving the stomach, small or large bowel, or rectum for conditions such as perforation, ischaemia, abdominal abscess, bleeding or obstruction.
► Washout/evacuation of intraperitoneal abdominal abscess (unless due to appendicitis or cholecystitis—excluded, see below).
► Washout/evacuation of intraperitoneal abdominal haematoma.
► Bowel resection/repair due to incarcerated umbilical, inguinal and femoral hernias (but not hernia repair without bowel resection/repair).
► Bowel resection/repair due to obstructing/incarcerated incisional hernias provided the presentation and findings were acute.
► Laparotomy or laparoscopy with inoperable pathology (ie, peritoneal/hepatic metastases).
► Laparoscopic/open adhesiolysis.
► Return to theatre for repair of substantial dehiscence of major abdominal wound (ie, 'burst abdomen') or any major postoperative complication (including all operations meeting the above criteria occurring as a complication of previous non-GI surgery, specific examples available at www.nela.org.uk/criteria).

### Exclusion criteria

► Frailty score not documented on preoperative admission clerking.
► Elective laparotomy/laparoscopy.
► Diagnostic laparoscopy/laparotomy where no further procedure is performed (N.B. if no procedure is performed because of inoperable pathology, then include).
► Appendicectomy +/-drainage of localised collection, unless the procedure is incidental to a non-elective procedure on the GI tract.
► Cholecystectomy +/-drainage of localised collection, unless the procedure is incidental to a non-elective procedure on the GI tract.
► (All surgery involving the appendix or gallbladder, including any surgery relating to complications such as abscess or bile leak is excluded. The only exception to this is if carried out as an incidental procedure to a more major procedure).
► Non-elective hernia repair without bowel resection.
► Minor abdominal wall dehiscence unless causing bowel complication requiring resection.
► Vascular surgery.

- ► Caesarean section or obstetric laparotomies.
- ► Gynaecological laparotomy (however bowel resection performed as non-elective procedure for obstruction due to cancer would be included).
- ► Ruptured ectopic pregnancy, or pelvic abscesses due to pelvic inflammatory disease.
- ► Laparotomy/laparoscopy for pathology caused by blunt or penetrating trauma.
- ► All surgery relating to organ transplantation (including returns to theatre for any reason following transplant surgery).
- ► Surgery relating to sclerosing peritonitis.
- ► Surgery for removal of dialysis catheters.
- ► Laparotomy/laparoscopy for oesophageal pathology.
- ► Laparotomy/laparoscopy for pathology of the spleen, renal tract, kidneys, liver, gall bladder and biliary tree, pancreas or urinary tract.

### Patient identification and data collection

Patients will be screened for inclusion criteria by the local team. Data collection will be carried out using the case report form presented in the online supplementary appendix A. Hospital or National Health Service (NHS) number will not be entered into this form but will be kept separately with a key sheet.

Basic demographics, comorbidities and polypharmacy data will be recorded. Comorbidities will be collected based on the Charlson Comorbidity Index, a validated measure of prognostic impact of multiple chronic illnesses.[11] This will allow for standardisation of comparisons between any groups. Data will also be collected on baseline independence status, assessed by the number of times social services provide care (1–4 times), and living in a residential or nursing home, measured both predischarge and postdischarge.

Frailty will be measured using the Clinical Frailty Score (online supplementary appendix B). This has been validated for use to assess frailty in older patients who underwent general surgery and OPSOC has successfully applied this before in previous work in this area.[12] The score ranks from 1 to 7 with a score of ≥5 being classed as frail.

Data will be collected on preoperative risk from scoring systems used commonly within emergency general surgery. This will include the P-POSSUM score[13] and the American Society of Anaesthesiologist grade.[14]

Data will be collected on operative procedures performed. Information will be obtained from patient case notes on 30-day outcomes. This includes 30-day mortality and evidence of postoperative complications. These complications will be rated using the Clavien-Dindo classification (online supplementary appendix C). This will allow for complications to be rated and outcomes to be assessed together. Finally, information will be obtained from the patient notes regarding 90-day mortality.

A detailed summary of the study questions asked can be found in the online supplementary appendix A.

The timetable for data collection is given in table 1.

**Table 1** Timetable for data collection

| Period | Date |
| --- | --- |
| Case identification period | 20/03/2017 - 19/06/2017 |
| Data collection completion date | 19/09/2017 |
| Validation completion date | 30/09/2017 |

### Primary outcome
90-day mortality

### Secondary outcomes
- ► Length of hospital stay (measured in days)
- ► Postoperative complications (yes/no and Clavien-Dindo grade of complication)
- ► Change in level of independence
- ► Length of stay on high dependency unit and intensive care unit (measured in days)
- ► Intermediate care stay on discharge (yes/no and duration of stay measured in length of days)
- ► 30-day mortality
- ► 30-day readmission

### Quality assurance
The study has been registered (www.clinicaltrials.gov, registration number NCT02952430).

The quality of this study has been assessed by the following means:
- ► Steering group meetings: 03/10/2016 and 13/12/2016.
- ► Review by OPSOC.
- ► Peer review by professionals with relevant expertise (Clinical trialists, statisticians, surgeons and geriatricians).
- ► Review by Research & Development department at NHS Greater Glasgow & Clyde (Sponsor Institution).
- ► Review by North West Surgical Trials Centre Trial Adoption Committee.

### Validation
Data validation will be performed by local teams on 25% of data fields for 10% of cases. The validated fields will include key demographic and outcome data.

### Data management
Completed datasets will be entered into an established and specifically designed online secure electronic database (REDCap, www.project-redcap.org). Password-protected login details will be provided to local collaborators permitting secure data entry into the database. All data will be handled in accordance with the Data Protection Act 1998. All transmission and storage of data will be encrypted and compliant with HIPAA security guidelines.

No patient identifiable information will be uploaded or stored on the secure database. Collaborators will anonymise patients by recording patient hospital numbers alongside database numbers in a separate spreadsheet in order to aid the collection of data locally.

## Statistical analysis and power calculation

Using OPSOC data, frailty exists in 28% of older patients admitted with emergency surgical conditions. Fifty-four per cent of the frail people who underwent surgery had died after 90 days. In order to detect a 10% difference in mortality rate at day 90 between frail and non-frail patients, a sample size of 480 is required, given an expected mortality proportion in those not frail of 0.075 and those frail of 0.175 (data from OPSOC), assuming an 80% power. We anticipate minimal patients who are lost to follow-up and to account for this, we will aim to recruit 500 patients.

Statistical support will be provided by OPSOC. Data will be analysed for correlation between frailty and postoperative outcomes, including 90-day mortality, complications and loss of independence.

Our primary analysis will be a logistic regression of 90-day mortality by frailty, adjusted for age (65–74, and >75 years old) and gender. We will carry out a secondary analysis of the primary outcome by including additional clinical mediators which are determined statistically important using a likelihood ratio test with a stepwise model fitting approach of nested regression models, and presented as a final multivariable model. All analyses will be presented as adjusted OR with associated 95% CIs and p values.

All other outcomes will be analysed as per the above analysis, but will be deemed secondary outcomes.

## Anticipated recruitment

Data will be collected at participating sites for all patients meeting the inclusion criteria over a 3-month period. This has been calculated based on information submitted by participating sites regarding the number of laparotomies performed per month on patients aged ≥65 years. According to this, 3 months should permit the identification of 500 patients.

## ETHICS AND DISSEMINATION
### Ethical approval

Ethical approval for this study was granted by a National Health Service Research Ethics Committee via the Proportionate Review Service. This was granted by the Black Country Research Committee on 28 November 2016 (REC Reference 16/WM/0500). The same committee reviewed the amended protocol and granted a favourable opinion on 6 February 2017.

## Registration

All participating units must obtain approval from their local Research & Development department consistent with the guidance from their relevant national organisation:. This study has been registered, reviewed and approved by the following organisations:

► The HRA (Health Research Authority) for sites in England

► The NRSPCC (NHS Research Scotland Permissions Co-ordinating Centre) for sites in Scotland

► The Health & Care Research Permissions Service for sites in Wales

The project will therefore be registered locally with the Trust Research & Development department prior to commencing patient identification and data collection at each site. It is the responsibility of the local ELF team to ensure that local Research and Development approvals are in place prior to commencing data collection.

## Dissemination

All data will be reported as a whole cohort. Unit level data for comparison will be fed back to collaborators to support local service improvement. This project will be submitted for presentation at a national or international surgical and geriatric conference. Manuscript(s) will be prepared following close of the project.

**Contributors** This is a collaborative publication. KLP, LP, JH and SM: conceived and developed the project. IF and JH: wrote the first draft of the manuscript. All authors: contributed to subsequent drafts of the manuscript.

**Competing interests** None declared.

**Provenance and peer review** Not commissioned; externally peer reviewed.

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
