## [Reviewer comments · BMJ Open]

ARTICLE DETAILS

TITLE (PROVISIONAL)	THE INFLUENCE OF FRAILITY IN OLDER PATIENTS UNDERGOING EMERGENCY LAPAROTOMY, A UK BASED OBSERVATIONAL STUDY
AUTHORS	Parmar, Kat; Pearce, Lyndsay; Farrell, Ian; Hewitt, Jonathan; Moug, Susan

VERSION 1 - REVIEW

REVIEWER	MR ANDREW RENWICK Royal Alexandra Hospital Corsebar Road Paisley Renfrewshire PA2 9PN
REVIEW RETURNED	08-Jun-2017

GENERAL COMMENTS	This is well written and an important piece of work which has relevance for surgeons the world over. The ELF study has generated a great amount of interest and support on social media and the authors are to be congratulated for what will be the first of many clinically relevant publications and a model which will be propagated by many others.
--

REVIEWER	Kjetil Søreide Stavanger University Hospital, Stavanger, Norway
REVIEW RETURNED	19-Jun-2017

GENERAL COMMENTS	This is a multicenter observation study to investigate the influence on frailty in emergency general surgery in patients aged 65 years and older. The study is important and may help inform on how frailty influences mortality in this particular group of aged patients. There are some issues that may be addressed. Intro: - This appears very UK centric and may be enhanced by addressing some more recent papers on the topic, see references listed below. Methods: - It should be clearly stated that the study intends to follow and report per the STROBE guidelines for observational studies. - Since inclusion is age >65 yrs, I would recommend to include appendectomies and cholecystectomies, as frequent procedures and with likely higher mortality than would be expected for these procedures in younger, elective patients. Exclusion of these may reduce the generalisability of the findings.
---

- The chosen frailty score is one of many, and why this score is chosen compared to others (see several references below) needs to be discussed. The chosen score is subjective and a more objective approach may be needed. The type of score or assessment of frailty in the emergency setting is not straight forward and this must be better recognised by the authors. How will you address this? Will you assess the validity of the score by test-retest for correlation and accuracy?

- Excluding surgery (lapscopy or laptomy) with non-resectional procedures may also be a bias, as again the elderly group who are subject to this in a "non-therapeutic" way indeed be at particular risk for mortality, so I would argue to include this group as well.

- The non-operated group (e.g. conservative managed conditions or those believed to be too frail to be offered surgery) is a particular problem as there is no way to address the selection in the study or to quantify the denominator of operated cases. Could you please comment on this?

Results:

- Not applicable at this stage

Discussion:

- There are clear limitations to the chosen frailty score, particularly since there are now some scores of frailty that have been tested and validated in the emergency surgery setting. It would be good to have any of these scores done in parallel to the chosen score or, at least, discuss this as a limitation. Again, the chosen references are not up to date on the topic, see list below.

References to include/consider:

Søreide K & Wijnhoven BP. Surgery for an ageing population. Br J Surg. 2016 Jan;103(2):e7-9. doi: 10.1002/bjs.10071. PubMed PMID: 26771471.

Desserud KF, et al. Emergency general surgery in the geriatric patient. Br J Surg. 2016 Jan;103(2):e52-61. doi: 10.1002/bjs.10044. PubMed PMID: 26620724.

Aunan JR, et al. Molecular and biological hallmarks of ageing. Br J Surg. 2016 Jan;103(2):e29-46. doi: 10.1002/bjs.10053. PubMed PMID: 26771470.

Søreide K & Desserud KF. Emergency surgery in the elderly: the balance between function, frailty, fatality and futility. Scand J Trauma Resusc Emerg Med. 2015 Feb 3;23:10. doi: 10.1186/s13049-015-0099-x. PubMed PMID: 25645443; PubMed Central PMCID: PMC4320594.

Orouji Jokar T, et al. Emergency general surgery specific frailty index: A validation study. J Trauma Acute Care Surg. 2016 Aug;81(2):254-60. doi: 10.1097/TA.0000000000001120. PubMed PMID: 27257694.

Joseph B, et al. Emergency General Surgery in the Elderly: Too Old or Too Frail? J Am Coll Surg. 2016 May;222(5):805-13. doi: 10.1016/j.jamcollsurg.2016.01.063. Epub 2016 Feb 26. PubMed

	PMID: 27113515. Kenig J, et al. Geriatric Assessment as a qualification element for elective and emergency cholecystectomy in older patients. World J Emerg Surg. 2016 Jul 29;11:36. doi: 10.1186/s13017-016-0094-1. eCollection 2016. PubMed PMID: 27478493; PubMed Central PMCID: PMC4966740. Wahl TS, et al. Association of the Modified Frailty Index With 30-Day Surgical Readmission. JAMA Surg. 2017 May 3. doi: 10.1001/jamasurg.2017.1025. PubMed PMID: 28467535.
--	---

VERSION 1 – AUTHOR RESPONSE

Reviewer: 1. Reviewer Name: MR ANDREW RENWICK.

Institution and Country: Royal Alexandra Hospital, Corsebar Road, Paisley, Renfrewshire, PA2 9PN.

Please state any competing interests: None Declared.

Please leave your comments for the authors below.

This is well written and an important piece of work which has relevance for surgeons the world over. The ELF study has generated a great amount of interest and support on social media and the authors are to be congratulated for what will be the first of many clinically relevant publications and a model, which will be propagated by many others.

Response Reviewer 1. No suggested changes by reviewer.

Reviewer: 2. Reviewer Name: Kjetil Søreide.

Institution and Country: Stavanger University Hospital, Stavanger, Norway

Please state any competing interests: None declared.

Please leave your comments for the authors below.

This is a multicenter observation study to investigate the influence on frailty in emergency general surgery in patients aged 65 years and older. The study is important and may help inform on how frailty influences mortality in this particular group of aged patients. There are some issues that may be addressed.

Intro:

- This appears very UK centric and may be enhanced by addressing some more recent papers on the topic, see references listed below.

Response: We agree that the paper is UK centric, which reflects that the two main collaborative groups are based in the U.K. (OPSOC and NWRC). In addition to this being one of the first studies to combine two established collaboratives, it is important to highlight the unique groups involved: medics and surgeons; trainees and consultants. As a result, it was restricted to the U.K. to take advantage of already established contacts and networks to optimise recruitment. We did receive interest from Europe regarding the study (Italy and Belgium), and would hope to include Europe in our future work. Regarding your suggested references we have read and reviewed these and have added several to our work.

Methods:

- It should be clearly stated that the study intends to follow and report per the STROBE guidelines for observational studies.

Response: We can confirm that the study will confirm to the STROBE guidelines on trial reporting for the full paper and will include a reference to these guidelines in our final paper.

- Since inclusion is age >65 yrs, I would recommend to include appendicectomies and cholecystectomies, as frequent procedures and with likely higher mortality than would be expected for these procedures in younger, elective patients. Exclusion of these may reduce the generalisability of the findings.

Response: There is a well-established National Database for emergency laparotomies in England and Wales, called NELA (National Emergency Laparotomy Audit). This audit includes all ages, but has excluded cholecystectomies and appendicectomies so as to not skew the mortality data, especially as many will be done laparoscopically as they would be in the older adult population. Our data collection is based on the inclusion/ exclusion criteria allowing ease of patient identification (as it is already being routinely collected) making this work simple and attractive for potential recruiting sites.

In addition, results from the most recent NELA report have shown that from 23k patients, mortality is highest in the over 65s and in those who survive, being over 65 results in a significantly longer hospital stay. In conclusion, we do not think exclusion of appendicectomy/ cholecystectomy will reduce the generalisability of our findings.

<http://www.nela.org.uk/reports> The Second Patient Report of the National Emergency Laparotomy Audit (NELA) December 2014 to November 2015.

- The chosen frailty score is one of many, and why this score is chosen compared to others (see several references below) needs to be discussed. The chosen score is subjective and a more objective approach may be needed. The type of score or assessment of frailty in the emergency

setting is not straight forward and this must be better recognised by the authors. How will you address this? Will you assess the validity of the score by test-retest for correlation and accuracy?

Response: We agree that the frailty score is one of many and will discuss in our final paper the reason for our selection with possible limitations of our chosen score. However, it is the one, which our OPSOC group has used previously in our published work in emergency surgical patients. As a result of our experience, we would argue that a simple 7-point frailty score is easy to use in an emergency situation and by clinicians who are not overly familiar with the concept of frailty. Further, while not specifically testing the frailty score, the study is due to undergo data validation at each site and where possible we will test for variation between sites.

- Excluding surgery (lapscopy or lapotomy) with non-resectional procedures may also be a bias, as again the elderly group who are subject to this in a "non-therapeutic" way indeed be at particular risk for mortality, so I would argue to include this group as well.

Response: We draw your attention to the earlier answer stating the reasons for our inclusion/exclusion criteria. Again, mortality was high in the over 65s from the results of the NELA work, highlighting how key the findings from ELF could potentially be in guiding future interventions in this vulnerable group.

- The non-operated group (e.g. conservative managed conditions or those believed to be too frail to be offered surgery) is a particular problem as there is no way to address the selection in the study or to quantify the denominator of operated cases. Could you please comment on this?

Response: The reviewer makes an excellent point. However, we would argue that we are trying to define if frailty influences mortality in older patients undergoing emergency laparotomy, not if frailty defines the operative decision. We know that mortality is high in the >65s and if frailty is an influencing factor then that opens up opportunities to address this by designing interventions. The influence that frailty may have on the operative decision is a separate study which although would provide valuable information, would be difficult to do. Indeed there is little published information on numbers that are declined emergency surgery and the reasons behind that.

Results:

- Not applicable at this stage.

Response: no questions are raised.

Discussion:

- There are clear limitations to the chosen frailty score, particularly since there are now some scores of frailty that have been tested and validated in the emergency surgery setting. It would be good to have any of these scores done in parallel to the chosen score or, at least, discuss this as a limitation. Again, the chosen references are not up to date on the topic, see list below.

References to include/consider:

Søreide K & Wijnhoven BP. Surgery for an ageing population. *Br J Surg*. 2016 Jan;103(2):e7-9. doi: 10.1002/bjs.10071. PubMed PMID: 26771471.

Desserud KF, et al. Emergency general surgery in the geriatric patient. *Br J Surg*. 2016 Jan;103(2):e52-61. doi: 10.1002/bjs.10044. PubMed PMID: 26620724.

Aunan JR, et al. Molecular and biological hallmarks of ageing. *Br J Surg*. 2016 Jan;103(2):e29-46. doi: 10.1002/bjs.10053. PubMed PMID: 26771470.

Søreide K & Desserud KF. Emergency surgery in the elderly: the balance between function, frailty, fatality and futility. *Scand J Trauma Resusc Emerg Med*. 2015 Feb 3;23:10. doi: 10.1186/s13049-015-0099-x. PubMed PMID: 25645443; PubMed Central PMCID: PMC4320594.

Orouji Jokar T, et al. Emergency general surgery specific frailty index: A validation study. *J Trauma Acute Care Surg*. 2016 Aug;81(2):254-60. doi: 10.1097/TA.0000000000001120. PubMed PMID: 27257694.

Joseph B, et al. Emergency General Surgery in the Elderly: Too Old or Too Frail? *J Am Coll Surg*. 2016 May;222(5):805-13. doi: 10.1016/j.jamcollsurg.2016.01.063. Epub 2016 Feb 26. PubMed PMID: 27113515.

Kenig J, et al. Geriatric Assessment as a qualification element for elective and emergency cholecystectomy in older patients. *World J Emerg Surg*. 2016 Jul 29;11:36. doi: 10.1186/s13017-016-0094-1. eCollection 2016. PubMed PMID: 27478493; PubMed Central PMCID: PMC4966740.

Wahl TS, et al. Association of the Modified Frailty Index With 30-Day Surgical Readmission. *JAMA Surg*. 2017 May 3. doi: 10.1001/jamasurg.2017.1025. PubMed PMID: 28467535.

Response: We feel that we have addressed these points in the responses above. To confirm, we will discuss the limitations of our chosen frailty score in the final paper.

VERSION 2 – REVIEW

REVIEWER	Kjetil Soreide Stavanger University Hospital, Norway
REVIEW RETURNED	30-Jul-2017

GENERAL COMMENTS	No further comments
---------------------